# Resiquimod-Mediated Activation of Plasmacytoid Dendritic Cells Is Amplified in Multiple Sclerosis

**DOI:** 10.3390/ijms20112811

**Published:** 2019-06-08

**Authors:** Marta Corsetti, Gabriella Ruocco, Serena Ruggieri, Claudio Gasperini, Luca Battistini, Elisabetta Volpe

**Affiliations:** 1Neuroimmunology Unit, IRCSS Fondazione Santa Lucia, 00143 Rome, Italy; m.corsetti@hsantalucia.it (M.C.); gabrie84@gmail.com (G.R.); l.battistini@hsantalucia.it (L.B.); 2Department of Neuroscience “Lancisi”, San Camillo Hospital, 00152 Rome, Italy; serena.ruggieri@gmail.com (S.R.); c.gasperini@libero.it (C.G.)

**Keywords:** multiple sclerosis, myeloid dendritic cells, plasmacytoid dendritic cells, costimulatory molecules, virus infection, Toll-like receptors, Resiquimod

## Abstract

Background: Multiple sclerosis (MS) is a chronic inflammatory autoimmune disease of the central nervous system. The cause of multiple sclerosis is unknown but there are several evidences that associate the genetic basis of the disease with environmental causes. An important association between viral infection and development of MS is clearly demonstrated. Viruses have a strong impact on innate immune cells. In particular, myeloid dendritic cells (mDCs) and plasmacytoid dendritic cells (pDCs), are able to respond to viruses and to activate the adaptive immune response. Methods: In this study we mimic viral infection using synthetic single-strand RNA, Resiquimod, and we compared the response of both DC subsets derived from healthy donors and MS patients by characterizing the expression of costimulatory molecules on the DC surface. Results: We found that pDCs from MS patients express higher levels of OX40-L, HLA-DR, and CD86 than healthy donors. Moreover, we found that blood cells from MS patients and healthy donors upon Resiquimod-stimulation are enriched in a subpopulation of pDCs, characterized by a high amount of costimulatory molecules. Conclusion: Overall, these results indicate that activation of pDCs is enhanced in MS, likely due to a latent viral infection, and that costimulatory molecules expressed on pDCs could mediate a protective response against the viral trigger of autoimmunity.

## 1. Introduction

Multiple sclerosis (MS) is an inflammatory, demyelinating disease of the central nervous system (CNS), and is the most common neurological cause of debilitation in adults, with an incidence of 0.1%. MS is currently believed to be an immune-mediated disorder where the immune response attacks the CNS and the pathology is directly mediated by autoantigen-specific T cells [1]. Cross-reactivity between nonself proteins, such as those from bacteria or viruses, and self proteins, termed molecular mimicry, has been proposed to be a possible mechanism for the onset of autoreactive T-cell responses [2,3,4]. In fact, the infectious etiology of MS has been suspected for a long time and by a large number of viruses: DNA viruses, such as the Epstein-Barr virus (EBV) and human herpesvirus 6 (HHV-6) [5], and RNA viruses, such as human endogenous retroviruses [6,7,8], and the coronavirus [9].

Dendritic cells (DCs) detect viral infection through Toll-like receptors (TLRs), key components of the innate immune system [10].

Once activated, DCs may trigger autoreactive T cells, with subsequent trafficking of activated T cells to the CNS [11]. However, it is likely that during a viral infection, DCs may also activate a protective immune response aimed to clear the viruses from the periphery and from the brain. The response of DCs to activating signals is a critical issue in determining the inflammatory or regulatory role of the immune response.

There are two subsets of peripheral blood DCs in humans, myeloid (mDCs) and plasmacytoid (pDCs), with distinct functions [12,13]. Previous studies on DCs and MS revealed that monocyte-derived DCs from patients produce higher levels of tumor necrosis factor-α (TNF-α) and interleukin-6 (IL-6) than healthy subjects [14]. In line with these observations, mDCs obtained from MS patients produce higher levels of IL-12 and IL-6 in response to a TLR-7/8 agonist [15]. Conversely, pDCs in MS patients produce lower levels of interferon (IFN)-α compared to healthy subjects [15]. Furthermore, it has been shown that upregulation of the costimulatory molecules upon stimulation with IL-3 and CD40L is significantly delayed in pDCs from MS patients compared to healthy donors [16]. The inefficient maturation of pDCs observed in this last paper could be associated with the specific stimulus. Indeed, the impaired expression of CD40 in pDCs from MS patients could impair the responsiveness to a CD40L stimulus [16]. Moreover, this study is not exhaustive because it is known that microbe-activated pDCs have features distinct from those of cytokine-activated DCs [17].

Overall, these studies indicate that DCs from MS patients and healthy donors respond differently to activating stimuli, suggesting that a better understanding of these differences may improve the knowledge on the mechanism regulating the balance between pro-inflammatory and anti-inflammatory immune responses in MS disease.

In our study, we hypothesized that there is a differential activation of DCs in response to viral infection in MS compared to healthy donors. In order to mimic a viral infection in mDCs and pDCs, we selected single-strand RNA Resiquimod, a strong signal able to simultaneously induce activation of both subsets of DC, pDCs and mDCs. We systematically compared the ex vivo expression of costimulatory molecules on mDCs and pDCs from MS patients and healthy donors. We found a stronger activation of pDCs from MS compared to healthy donors in response to Resiquimod, that could be a mechanism of defense acquired during the disease, and that could be further therapeutically amplified to increase the antiviral response in MS.

## 2. Results

### 2.1. Differential Activation of Myeloid and Plasmacytoid Dendritic Cells in MS Patients and Healthy Donors

In order to analyze the activation of DCs in MS patients and healthy donors, we performed ex vivo cultures of peripheral blood mononuclear cells (PBMC) from blood of MS patients and healthy donors, that we stimulated with either Resiquimod or medium alone. We discriminated the mDC population among PBMC using multicolor flow cytometry (Figure 1a), and we analyzed their activation using fluorescent antibodies against costimulatory molecules. In particular, we focused our attention on three major costimulatory molecules expressed on the DC surface known to activate T cells: HLA-DR, CD86, and OX40-L. We found that all activation markers were increased in Resiquimod-stimulated mDCs (Figure 1b,c) obtained from both groups of individuals compared to unstimulated cells. However, mDCs from MS patients expressed similar levels of those molecules compared to mDCs from healthy donors (Figure 1b,c).

Similarly, we analyzed the activation of pDCs using antibodies against markers enabling their discrimination, such as CD123 (Figure 2a). Although Resiquimod induces upregulation of all activation markers in both MS patients and healthy donors, pDCs from MS patients compared to pDCs from healthy donors expressed higher levels of OX40-L and HLA-DR upon stimulation (Figure 2b,c). Moreover, expression levels of CD86 in pDCs from MS patients were already higher in the unstimulated samples (Figure 2b,c).

### 2.2. OX40L Regulates IL-9 Production by Human CD4 T Lymphocytes

Given the IL-9 induction by purified pDCs demonstrated in a previous study [18], we hypothesized that OX40L overexpressed by pDCs from MS patients is involved in stimulating IL-9 by naïve CD4 T cells differentiated in Th cells. In order to test this hypothesis, we performed an experiment where we stimulated purified naïve CD4 T cells with different doses of recombinant OX40L (Figure 3a). Interestingly, high dose of OX40L induce IL-9 production, indicating that OX40L overexpression by pDCs of MS patients may contribute to the induction of a T helper 9 (Th9) response. Moreover, recombinant OX40L further increases IL-9 production by Th9 cells obtained by stimulating naïve CD4 T cells with TGF-β and IL-4 (Figure 3b).

### 2.3. pDC Subpopulations in MS Patients Compared to Healthy Donors

Activation-induced pDCs can be differentiated in three subpopulations, recently identified on the basis of the expression of the immune-checkpoint molecules: CD80 and PDL-1 [19]. In particular P3 subpopulation expresses high amount of the costimulatory molecules, including CD80, CD86 and OX40-L. In order to investigate whether unstimulated or Resiquimod-stimulated pDCs from MS patients were enriched in the P3 subpopulation, we measured the frequency of PDL-1^+^ CD80^−^ (P1), PDL-1^+^ CD80^+^ (P2), and PDL-1^−^ CD80^+^ (P3) within PBMC in both healthy donors and MS patients. Although the ex-vivo expression of CD80 and PDL-1 on unstimulated pDCs is very low, the frequency of the P3 subpopulation is significantly higher than P1 and P2 (Figure 4a,b). However, the three subpopulations increase significantly upon stimulation with Resiquimod, in both healthy donors and MS patients (Figure 4a,b), as previously demonstrated [19]. Moreover, we observed that in healthy donors the frequencies of P1 and P3 are significantly higher compared to the P2 subpopulation, while in MS patients the frequency of P3 is significantly higher than P1 and P2, whose frequencies are similar (Figure 4b). As expected, the evaluation of the median fluorescence intensity (MFI) of CD80 and PDL-1 reflects the frequency of the three pDC subpopulations. In fact, we found a significant induction of both markers on pDCs upon stimulation. However, no differences between healthy donors and MS patients were detected (Figure 4c).

## 3. Discussion

Our results indicate that pDCs from MS patients are differentially activated compared to pDCs from healthy donors in response to single strand RNA mimicked by Resiquimod. Interestingly, mDCs from the same individuals and activated with the same stimulus behave similarly in MS and healthy donors. This could be explained by the downstream signaling activated by TLR-7 and TLR-8 in pDCs and mDCs, respectively [20]. Moreover, fast indirect signaling pathways could also influence pDC activation. However, these results suggest a potential differential induction of T cell immune response induced by pDCs in MS patients compared to healthy donors. For instance, it is known that a TLR-7 agonist induces lytic properties of pDCs via the tumor necrosis factor-related apoptosis-inducing ligand (TRAIL), whose expression is directly correlated with individual viral loads in HIV+ patients [21]. Similarly, pDCs express granzyme B, which mediates killing of CD4 T cell during viral infections like HIV [22,23]. Moreover, we reported that pDCs from MS patients stimulated with Resiquimod and co-cultured with naïve T cells significantly induce the secretion of IL-9, a cytokine produced by a new Th profile called Th9 [18]. Interestingly, in a mouse model it has been demonstrated that OX40-L, a costimulatory molecule that we found highly expressed in pDCs from MS patients, contributes to IL-9 polarization [24]. Here, we found that recombinant OX40L induce and amplify IL-9 production by human T cells, indicating that the interaction OX40-OX40-L between pDCs and T cells in MS patients could be responsible of the IL-9 induction observed in MS patients. Other costimulatory molecules upregulated in pDCs from MS patients, such as HLA-DR and CD86, are considered as the predominant molecules delivering ”first and second signals” during T-cell activation [25]. Thus, our results indicate that OX40-L could be the crucial costimulatory molecule determining the fate of the immune response generated by pDCs. In fact, OX40 and OX40-L dictate the different type of effector (protective or pathogenic) T cells that accumulate in primary and secondary responses, as well as determine the frequency of memory T cells that are generated [26]. Ligation of OX40 can induce Tfh polarization [27,28], an inflammatory Th2 response [29], Th1 and Th2 [30,31], Th9 [24], Th17 [32], and T regulatory cells [33], depending on the context. Although OX40^+^ lymphocytes have been identified within active lesions of MS brain tissues [34], the role of the OX40-OX40-L axis in MS is not elucidated.

Moreover, OX40 signals can mediate CD8 immune responses, which is particularly relevant during viral infection. It has been demonstrated that the OX40-OX40-L axis can indirectly promote CD8 T cell priming through augmenting CD4 T cell help [35], and directly enhances the expansion of memory CD8 T cell populations specific for viruses [36], including EBV, which has been widely associated with MS [37,38]. In MS, the frequency of CD8^+^ T cells specific for EBV lytic and latent antigens is higher in active and inactive MS patients, respectively, compared to healthy donors, underlying the importance of the activation of the anti-EBV response in MS [39]. In line with the effect of the OX40-OX40-L axis in CD8 T cell activation, OX40 agonistic therapy has proven a potent cancer immunotherapy, enhancing CD8 T cell tumor infiltration [40]. Moreover, our analysis indicated that the frequency of pDC subpopulations in the blood of MS patients does not differ significantly from that of healthy donors, according to the similar levels of CD80 and PDL-1 in pDCs from MS patients and healthy donors. However, the P1 subpopulation of pDCs has been associated to autoimmune diseases linked to pDC-derived type I interferon (IFN) [19], such as psoriasis and lupus erythematous (SLE) [41,42,43]. However, the physiopathology of MS differs from SLE and psoriasis [44]. In fact, the majority of MS patients benefits from IFN-β therapy [45]; that is, pDCs from MS patients produce lower levels of IFN-α compared to those derived by healthy donors [15], and the serum levels of type I IFN as well as the response of PBMC to type I IFN in MS are lower compared to SLE patients [46].

This study highlights the potential role of pDCs and the OX40-O40L axis in the protective antiviral response in MS. This mechanism could contribute to the balance between inflammation and immune regulation that could be therapeutically modulated in MS patients towards a protective immune response.

## 4. Materials and Methods

### 4.1. MS Subjects

Patients with a relapsing–remitting (RR)-MS inactive state according to established criteria [47] were enrolled in the study. All patients included in this study did not take immunomodulant or immunosuppressive compounds at least 2 months before recruitment. Approval by the ethics committee of the San Camillo Hospital (224/2015; 5 November 2015) and written informed consent in accordance with the Declaration of Helsinki from all participants were obtained before study initiation.

As controls, we used blood from age- and gender-matched individuals without inflammatory or degenerative diseases of the central or peripheral nervous system. These subjects were volunteers that underwent blood drawing.

### 4.2. Purification and Stimulation of Peripheral Blood Mononuclear Cells from Adult Blood

Peripheral blood mononuclear cells (PBMC) were isolated by Ficoll gradient centrifugation (GE Healthcare, Little Chalfont, UK) from whole blood. PBMCs were cultured at a density of 1 × 10^6^ cells/mL in 24-well plates (Corning, New York City, NY, USA) in RPMI 1640 with 10% of Fetal Bovine Serum (FBS), in the presence or absence of Resiquimod (R848) (Invivogen, San Diego, CA, USA) (100 ng/mL) for 6 hours at 37 °C. Then, cells were collected and stained with specific antibodies discriminating viable pDCs and mDCs and evaluating their expression of activation markers. After the isolation, the cells were stained with specific antibodies enabling to discriminate mDCs and pDCs.

### 4.3. Purification and Stimulation of Naïve CD4 T Cells

PBMC were stained with the anti-CD4-FITC (Miltenyi; 1/100), and CD4^+^ T Lymphocytes were purified by immunomagnetic selection, using the anti-FITC isolation kit (Miltenyi, Bergisch Gladbach, Germany). After the isolation, the cells were stained with anti-CD4 FITC (Miltenyi; 1/100), anti-CD45RA BV421 (BD Biosciences, San Jose, CA, USA; 1/60), anti-CD45RO PE (BD Biosciences; 1/30), anti-CD27 APC (Miltenyi; 1/60), and CD4^+^ naive T-cells were sorted by a MoFlo high speed cell sorter (Beckman Coulter, Atlanta, GA, USA) as CD4^high^, CD45RA^high^, CD45RO^−^, and CD27^+^. Sorted cells had a purity of over 97%, measured by flow cytometry (data not shown). Naive CD4^+^ T-cells were cultured in 96-well round-bottomed plates (Corning) at a density of 5 × 10^4^ per well in X-VIVO 15 serum-free medium (Lonza, Walkersville, MD, USA) in the presence of a Dynabeads CD3-CD28 T-cell expander (one bead per cell; Life Technologies, Carlsbad, CA, USA). TGF-β (5 ng/mL), IL-4 (50 ng/mL) (Miltenyi), and OX40L (RnD, Minneapolis, MN, USA) were used to stimulate naïve CD4 T cells. After 6 days, supernatants were collected for IL-9 quantification.

### 4.4. Analysis of Cytokine Production

IL-9 in culture supernatants was measured by IL-9 ELISA (Life Technologies) according to the manufacturer’s instructions.

### 4.5. Flow Cytometry Analysis

PBMC were harvested after 6 hours of culture and resuspended in an EDTA-containing medium, then stained for 15 minutes at 4 degrees with the following antibodies: Anti-human CD45-FITC (Coulter; 1/200), OX40-L-PE (BD; 1/50), CD4-ECD (Coulter; 1/100), CD16-BB700 (Pharmigen; 1/100), PDL-1-PC7 (Coulter; 1/100), CD86-APC (Miltenyi; 1/80), CD11c-APC-AlexaFluor700 (Coulter; 1/100), CD14-APC-Vio770 (Miltenyi; 1/120), CD3-e450 (eBioscience; 1/160), CD19-PB (Dako; Santa Clara, CA, USA 1/60), CD56-BV421 (BD; 1/50), Fixable Aqua Dead Cell stain (Thermofisher; 1/200), CD123-BV605 (Biolegend, San Diego, CA, USA; 1/30), CD80-BV650 (BD; 1/50), and HLA-DR-BV785 (BD; 1/100). Samples were washed in EDTA-containing medium, acquired using a Cytoflex cytometer (Coulter) and analyzed using FlowJo-10 software (version 10.3.0).

### 4.6. Statistical Analysis

For pair-wise comparisons of different conditions from the same donors or different donors, we used a non-parametric two-tailed paired or unpaired t-test, respectively. Data were presented as the mean ± standard error (SEM). The *p* values (*p*) of 0.05 or less were considered statistically significant.

## Figures and Tables

**Figure 1 ijms-20-02811-f001:**
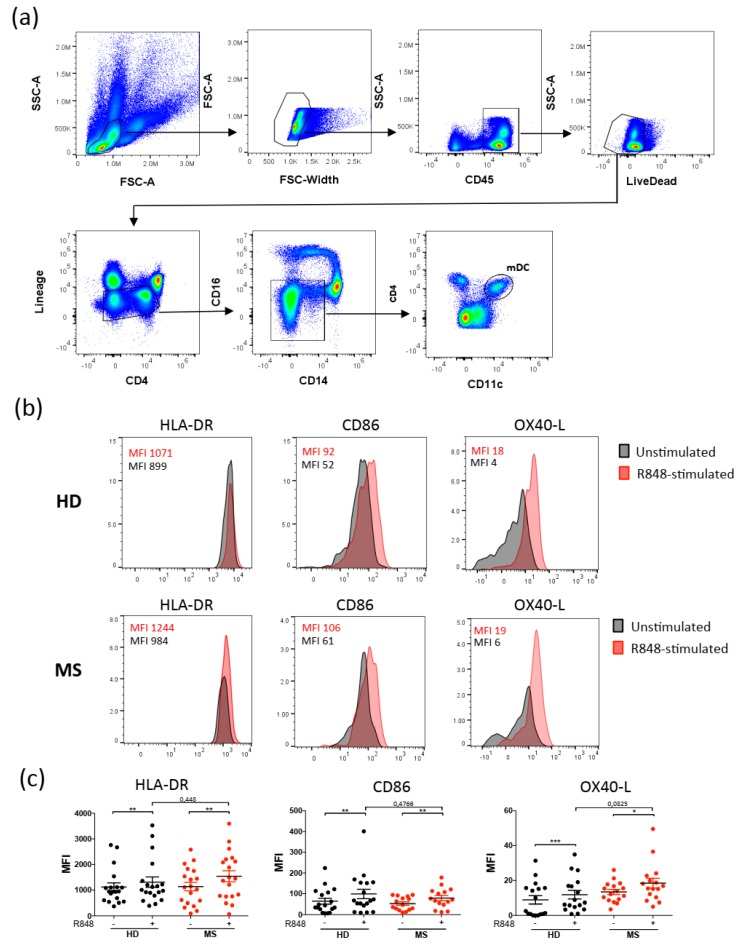
Similar activation of myeloid dendritic cells in multiple sclerosis (MS) patients and healthy donors. Peripheral blood mononuclear cells (PBMC) from healthy donors or MS patients were stimulated with Resiquimod (R848) or medium alone for 6 hours, then stained with antibodies against CD3, CD19, CD56 (Lineage), CD45, CD14, CD16, CD4, and Fixable Aqua Dead Cell stain. Viable myeloid dendritic cells (mDCs) are discriminated as singlets, CD45^+^, live dead^−^, Lineage^−^, CD4^+^, CD14^−^, CD16^−^, and CD11c^+^ (**a**). Further staining with antibodies against HLA-DR, CD86, and OX40-L enabled evaluation of mDC activation. Flow cytometry analysis of a representative experiment is shown in (**b**). Graphs represent the results of independent experiments (**c**). (* *p* < 0.05; ** *p* < 0.01; *** *p* < 0.001).

**Figure 2 ijms-20-02811-f002:**
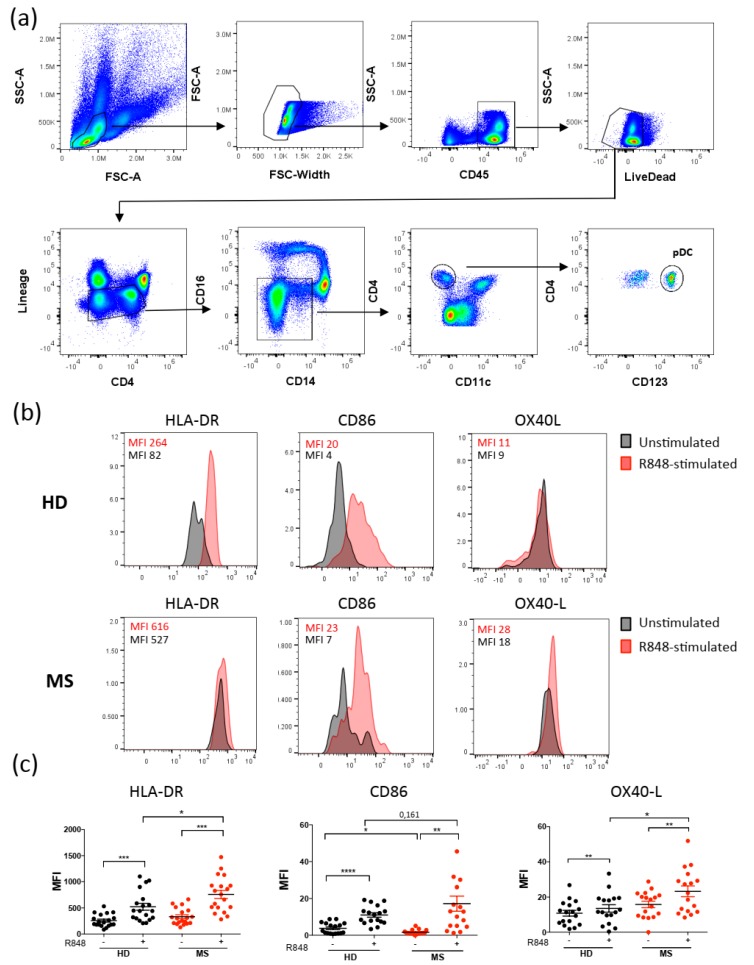
Activation of plasmacytoid dendritic cells is increased in MS patients versus healthy donors. Peripheral blood mononuclear cells from healthy donors or MS patients were stimulated with Resiquimod (R848) or medium alone for 6 hours, then stained with antibodies against CD3, CD19, CD56 (Lineage), CD45, CD14, CD16, CD4, and Fixable Aqua Dead Cell stain. Viable plasmacytoid dendritic cells (pDCs) are discriminated as singlets, CD45^+^, live dead^−^, Lineage^−^, CD4^+^, CD14^−^, CD16^−^, CD11c^−^, and CD123^+^ (**a**). Further staining with antibodies against HLA-DR, CD86, and OX40-L enabled evaluation of pDC activation. Flow cytometry analysis of a representative experiment is shown in (**b**). Graphs represent the results of independent experiments (**c**). (* *p* < 0.05; ** *p* < 0.01; *** *p* < 0.001; **** *p* < 0.0001).

**Figure 3 ijms-20-02811-f003:**
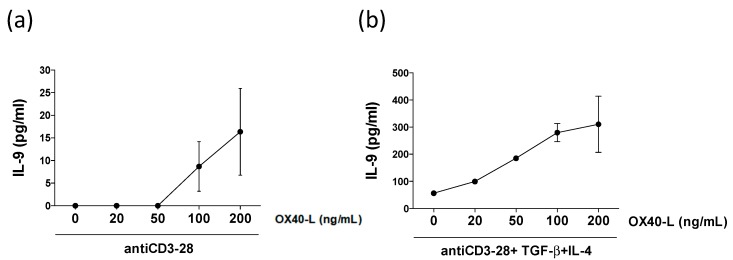
OX40L regulates IL-9 production by human CD4 T lymphocytes. Naive CD4 T cells were stimulated with anti-CD3/28 coated beads in absence of cytokines (**a**) and in presence of T helper 9 polarizing conditions (TGF-β + IL-4) (**b**), and indicated doses of recombinant OX40L for 6 days. Protein expression of IL-9 in the supernatants was analyzed by ELISA. Graphs represent the results of three independent experiments on three healthy donors.

**Figure 4 ijms-20-02811-f004:**
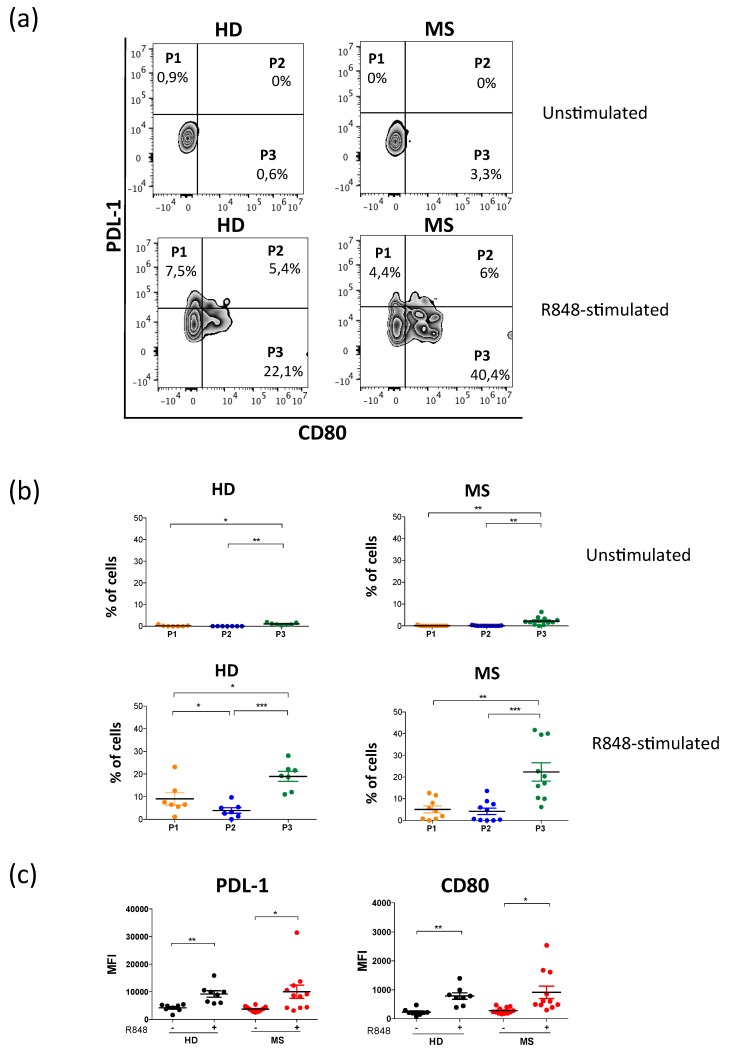
The P3-pDC subpopulation is enriched in MS compared with the P1 and P2 populations. Peripheral blood mononuclear cells from healthy donors or MS patients were stimulated with Resiquimod (R848) or medium alone for 6 hours, then stained with antibodies against CD3, CD19, CD56 (Lineage), CD45, CD14, CD16, CD4, and Fixable Aqua Dead Cell stain to discriminate viable pDCs, CD80, and PDL-1 to discriminate P1, P2, and P3 populations. Flow cytometry analysis of a representative experiment is shown in (**a**). Graphs represent the frequency of P1, P2, and P3 populations (**b**), or median fluorescence intensity of PDL-1 and CD80 (**c**) of independent experiments. (* *p* < 0.05; ** *p* < 0.01; *** *p* < 0.001).

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
