# Peer review of "Resiquimod-Mediated Activation of Plasmacytoid Dendritic Cells Is Amplified in Multiple Sclerosis"

_ijms, 2019, doi:10.3390/ijms20112811_

Round 1
Reviewer 1 Report
This interesting study is based on the idea that dendritic cells (DCs) from MS patients are differentially activated in response to viral infection. The authors investigated this problem by subjecting myeloid (mDC) and plasmocytoid (pDC) DCs to a simulated viral infection using ex-vivo cultures and single strand RNA Resiquimod. Comparison to cells from healthy individuals indicated a stronger response from the pDCs from MS patients, suggesting a differential pDC-dependent induction of T cell immune response. There is no doubt that the results are very interesting and highly relevant to the understanding and development of treatments for MS.
The experiments are well designed and the manuscript is well written.
However, some important details appear to be missing in the text.
- The Material and Methods section does not indicate the ex-vivo culture conditions and incubation medium.
- Likewise, information appears to be missing regarding the commercial origin of Resiquimod and the concentration at which the drug was used.
Author Response
Response to Reviewer 1:
Point 1: This interesting study is based on the idea that dendritic cells (DCs) from MS patients are differentially activated in response to viral infection. The authors investigated this problem by subjecting myeloid (mDC) and plasmocytoid (pDC) DCs to a simulated viral infection using ex-vivo cultures and single strand RNA Resiquimod. Comparison to cells from healthy individuals indicated a stronger response from the pDCs from MS patients, suggesting a differential pDC-dependent induction of T cell immune response. There is no doubt that the results are very interesting and highly relevant to the understanding and development of treatments for MS. The experiments are well designed and the manuscript is well written. However, some important details appear to be missing in the text. The Material and Methods section does not indicate the ex-vivo culture conditions and incubation medium. Likewise, information appears to be missing regarding the commercial origin of Resiquimod and the concentration at which the drug was used.
Response 1: We apologized for the negligence to have not well defined the ex-vivo culture conditions, commercial origin of Resiquimod and concentration. Accordingly, we included in the revised manuscript (material and methods section, line 198-201 all the details.

Reviewer 2 Report
Corsetti M et al., demonstrate that the pDCs in MS patients are already in an activated state and the response is amplified upon resiquimod exposure. The piece of data is interesting and the experiments are performed with multiple number of patients samples.
There is a high chance of pDCs are getting exposed to the self-nucleotides from the dying cells in MS patients, specifically in MS plaque and acquire an activated phenotype. One of major drawback in studies where people compare the functional outcome of mDC and pDCs from MS patients and attribute the outcome to a specific subset like pDC, may be a biased conclusion and does not consider other factors into account. For eg :- the DC subsets have different TLRs expression profile, mDCs have TLR8 and pDCs have TLR7, the response will be different for the same TLR agonist due to the nature of downstream signaling, sensitivity and the difference in the cytokines secreted by the DCs. R848 may be an appropriated TLR ligand specifically supporting the pDCs to acquire the mature phenotype to generate IL-9 producing CD4 T cells. Another TLR ligand like Poly I:C may do the same for mDCs and acquire the same functional outcome. As far as I understand from the protocol, the total PBMNCs were stimulated with R848 and it can be an indirect effect and the other cell subset in the PBMNCs get activated and further activate pDCs but may have a inhibitory effect on mDCs. ( for eg : CXCL4 can increase the IFN production , https://www.ncbi.nlm.nih.gov/pubmed/31043596 ) This is generally noted that the pDCs get activated better in bulk PBMNC than isolated from PBMNC, when activated with R848. I am wondering how authors are ruling out those possibilities and discrepancies?
There are reports on the role of TRAIL (PMID: 19690337) or GZMB mediated killing of CD4 T cell by pDCs during other viral infections like HIV ( PMID: 19965634, PMID: 23785122) or even on EAE models or type I diabetes , the role of pDCs are demonstrated as a critical cell subset for induction of tolerance . Also the OX40-OX40L axis mediated CD8 T cells can be more damaging to the neurons where CD8 T cells attack the neurons than protecting from any viruses.
1. The main data in the article is the surface staining of DC subsets under R848 activation and it will be great to see at least one functional data to support any of your conclusions included in the discussions based on the previous publication. At least demonstrate the IFNα or other cytokines in the culture supernatant of PBMNCS from healthy donors compared to MS patients to demonstrate that they are really different.
Other minor comments
1. Figure 1 C: Please show the statistical difference between the HD vs MS upon R848 stimulation as you depicted in Figure 2C. Visually there is difference in the HLADR and OX40L expressions.
2. What is the dose of R848 used for the stimulation and include the procedure for cell activation, FACS staining protocol – incubation time or any other specific methods used
Author Response
Response to Reviewer 2:
Point 1: Corsetti M et al., demonstrate that the pDCs in MS patients are already in an activated state and the response is amplified upon resiquimod exposure. The piece of data is interesting and the experiments are performed with multiple number of patients samples. There is a high chance of pDCs are getting exposed to the self-nucleotides from the dying cells in MS patients, specifically in MS plaque and acquire an activated phenotype. One of major drawback in studies where people compare the functional outcome of mDC and pDCs from MS patients and attribute the outcome to a specific subset like pDC, may be a biased conclusion and does not consider other factors into account. For eg :- the DC subsets have different TLRs expression profile, mDCs have TLR8 and pDCs have TLR7, the response will be different for the same TLR agonist due to the nature of downstream signaling, sensitivity and the difference in the cytokines secreted by the DCs. R848 may be an appropriated TLR ligand specifically supporting the pDCs to acquire the mature phenotype to generate IL-9 producing CD4 T cells. Another TLR ligand like Poly I:C may do the same for mDCs and acquire the same functional outcome.
Response 1: We agree with reviewer that Resiquimod on pDCs and mDCs activates different downstream signalling, that could explain the differential expression compared to MS observed in pDCs and not in mDCs. We discussed this point in the revised version of the manuscript (Line 139-140). We also appreciated the reviewer comments about the possibility that mDCs stimulated with another viral TLR ligand like Poly I:C may acquire a different phenotype. However, we selected Resiquimod as stimulus that can simultaneously activates both DC subsets (as reported at line 63-65 of the revised manuscript), and we found interesting that the same trigger (TLR7/8 agonist) activates both pDCs and mDCs but at different level in MS and HDs. These results could have relevance in a specific context where a single-strand RNA, mimicked by Resiquimod, encounters both DC subsets in vivo.
Point 2: As far as I understand from the protocol, the total PBMNCs were stimulated with R848 and it can be an indirect effect and the other cell subset in the PBMNCs get activated and further activate pDCs but may have a inhibitory effect on mDCs. ( for eg : CXCL4 can increase the IFN production , https://www.ncbi.nlm.nih.gov/pubmed/31043596 ) This is generally noted that the pDCs get activated better in bulk PBMNC than isolated from PBMNC, when activated with R848. I am wondering how authors are ruling out those possibilities and discrepancies?
Response 2: We thank the reviewer and we understand his concerns about the stimulation of total peripheral blood mononuclear cells. We agree with reviewer that using this experimental approach we cannot definitively conclude that we are looking a direct effect, although the short time of incubation (6hours) limits indirect effects. On the other side, this approach simulates the physiologic in vivo interaction between TLR ligand and TLR receptor in a complex system that include direct and indirect effects on immune cells. However, we added a sentence (Line 140-141) in the revised version of the manuscript to discuss this point.
Point 3: There are reports on the role of TRAIL (PMID: 19690337) or GZMB mediated killing of CD4 T cell by pDCs during other viral infections like HIV ( PMID: 19965634, PMID: 23785122) or even on EAE models or type I diabetes , the role of pDCs are demonstrated as a critical cell subset for induction of tolerance . Also the OX40-OX40L axis mediated CD8 T cells can be more damaging to the neurons where CD8 T cells attack the neurons than protecting from any viruses.
The main data in the article is the surface staining of DC subsets under R848 activation and it will be great to see at least one functional data to support any of your conclusions included in the discussions based on the previous publication. At least demonstrate the IFNα or other cytokines in the culture supernatant of PBMNCS from healthy donors compared to MS patients to demonstrate that they are really different.
Response 3: It has been demonstrated that in similar experimental conditions human pDCs from MS patients produce lower levels of interferon (IFN)- compared to those from healthy subjects, cited at Line 50-51.
We are particularly interested to IL-9 induction by purified pDCs, demonstrated in a previous study (Ruocco et al., Clin.Science 2015). In fact, we hypothesised that OX40L overexpressed by pDCs from MS patients, is involved in stimulating IL-9 by naïve CD4 T cells differentiated in Th cells. In order to test this hypothesis we performed an experiment where we stimulated purified naïve CD4 T cells with different doses of recombinant OX40L (Figure for Reviewer, red line). Interestingly, high dose of OX40L induce IL-9 production, indicating that OX40L overexpression by pDCs of MS patients may contribute to the induction of a T helper 9 (Th9) response. Moreover, recombinant OX40L further increases IL-9 production by Th9 cells obtained by stimulating naïve CD4 T cells with TGF-b and IL-4 (Figure for Reviewer, blue line).
Concerning other functions, at the moment we did not investigate the expression of TRAIL or GZMB that could mediate killing T cell during viral infections. However, we cited these important references in the revised version of the manuscript where we discuss these other potential roles of pDCs (Line 142-146).
Point 4: Figure 1 C: Please show the statistical difference between the HD vs MS upon R848 stimulation as you depicted in Figure 2C. Visually there is difference in the HLADR and OX40L expressions.
Response 4: The expression of HLA-DR and OX40L in stimulated mDCs of HD and MS are not statistically different. Since the reviewer found that visually there is a difference, we added “ns” in Figure 1c to clarify and avoid confusion.
Point 5: What is the dose of R848 used for the stimulation and include the procedure for cell activation, FACS staining protocol – incubation time or any other specific methods used
Response 5: We apologized for the negligence to have not well defined the concentration of Resiquimod, procedure for cell activation, FACS staining protocol and other specific methods. Accordingly, we included in the revised manuscript all the details (material and methods section, line 198-201; 204-205; 211).

Round 2
Reviewer 2 Report
The authors made a really good attempt for answering the reviewers concern and few minor additions are necessary to finalize the manuscript. The suggestions are listed below,
The authors are proposing the use of R848 as model for single stranded RNA and extrapolate the model to represent the viral infection in MS. Authors themselves associate the MS with EBV and HHV-6, both are DNA virus. The role of pDCs sensing DNA from dying cells in many autoimmune disease are well documented and may be that model is more suitable for a MS in the presence of a DNA virus. The concept of a single stranded RNA virus in MS and use of R848 for modeling the disease may not be very convincing or in other words your introduction of a virus association with MS and the model used to conclude the hypothesis does not matching. I think there are other reports on single-stranded RNA viruses ( eg:- coronavirus) on MS and authors should make some modifications in the literature to fit their hypothesis and experimental models.
Line 37 : HHV-6 – Human herpes virus – not just herpes virus
Figure 1: please show the p Values instead of non-significant
Please add the IL-9 secretion figure to the manuscript to support your concept and hypothesis. Please give the experimental details, number replicates and please include at least 3 donors.
thank you very much
Author Response
Response to Reviewer:
The authors made a really good attempt for answering the reviewers concern and few minor additions are necessary to finalize the manuscript. The suggestions are listed below,
The authors are proposing the use of R848 as model for single stranded RNA and extrapolate the model to represent the viral infection in MS. Authors themselves associate the MS with EBV and HHV-6, both are DNA virus. The role of pDCs sensing DNA from dying cells in many autoimmune disease are well documented and may be that model is more suitable for a MS in the presence of a DNA virus. The concept of a single stranded RNA virus in MS and use of R848 for modeling the disease may not be very convincing or in other words your introduction of a virus association with MS and the model used to conclude the hypothesis does not matching. I think there are other reports on single-stranded RNA viruses ( eg:- coronavirus) on MS and authors should make some modifications in the literature to fit their hypothesis and experimental models.
We thank reviewer for this important comment. We modified introduction, according to his suggestion (line 36-39).
Line 37 : HHV-6 – Human herpes virus – not just herpes virus
We corrected the mistake (Line 37 and list of abbreviations)
Figure 1: please show the p Values instead of non-significant
We modified figures showing the p values instead of non-significant.
Please add the IL-9 secretion figure to the manuscript to support your concept and hypothesis. Please give the experimental details, number replicates and please include at least 3 donors.
We added IL-9 secretion figure in the manuscript (Figure 3a,b). We described the figure (line 112-120), we wrote experimental details and number replicates (line 122-125; line 221-235), and we discussed these results (line 168-171).
